# Prevention and Treatment of Grade C Postoperative Pancreatic Fistula

**DOI:** 10.3390/jcm11247516

**Published:** 2022-12-19

**Authors:** Chengzhi Xiang, Yonghua Chen, Xubao Liu, Zhenjiang Zheng, Haoqi Zhang, Chunlu Tan

**Affiliations:** Department of Pancreatic Surgery, West China Hospital, Sichuan University, Chengdu 610041, China

**Keywords:** grade C postoperative pancreatic fistula, infection, bacterial translocation, operation techniques, treatment

## Abstract

Postoperative pancreatic fistula (POPF) is a troublesome complication after pancreatic surgeries, and grade C POPF is the most serious situation among pancreatic fistulas. At present, the incidence of grade C POPF varies from less than 1% to greater than 9%, with an extremely high postoperative mortality rate of 25.7%. The patients with grade C POPF finally undergo surgery with a poor prognosis after various failed conservative treatments. Although various surgical and perioperative attempts have been made to reduce the incidence of grade C POPF, the rates of this costly complication have not been significantly diminished. Hearteningly, several related studies have found that intra-abdominal infection from intestinal flora could promote the development of grade C POPF, which would help physicians to better prevent this complication. In this review, we briefly introduced the definition and relevant risk factors for grade C POPF. Moreover, this review discusses the two main pathways, direct intestinal juice spillover and bacterial translocation, by which intestinal microbes enter the abdominal cavity. Based on the abovementioned theory, we summarize the operation techniques and perioperative management of grade C POPF and discuss novel methods and surgical treatments to reverse this dilemma.

## 1. Introduction

Postoperative pancreatic fistula is the most annoying postoperative complication after pancreatic resection and is the main predicament faced by pancreatic surgeons. Although the mortality rates after pancreaticoduodenectomy in large-capacity institutions have dropped to less than 5% due to significant progress in surgical techniques and perioperative management [1], the incidence of pancreatic fistulas remains high, ranging between 3% and 45% [2]. Furthermore, POPF can lead to a series of secondary injuries, such as delayed gastric emptying, abdominal cavity infection, false aneurysms, and abdominal bleeding [3]. Of all the complications, the postoperative hemorrhage of the pancreas (PPH) is the most fatal, occurring in from 3% to 20% of patients, but the associated mortality is as high as 20% to 50% [4,5].

To standardize the definition of pancreatic fistula in different regions, the International Study Group on Pancreatic Fistula (ISGPF) published the first objective definition in 2005 [6] and modified it in 2016 [7]. Pancreatic fistula is divided into biochemical fistula and clinically significant POPF by the ISGPF’s definition. Grade A POPF is newly defined as a biochemical leak because it is considered to have no clinical impact on the normal postoperative course. Clinically significant POPFs are classified as grade B and grade C. Grade B has clinical influence and requires a change in management, including therapeutic agents and less-invasive treatment or percutaneous, endoscopic, or angiographic interventional procedures. Finally, grade C fistulas are defined as those that lead to reoperation, single or multiple organ failure, or even death [7,8].

Based on this definition, the incidence of clinically significant POPF has been reported to be approximately 1% to 36%, and the incidence of grade C POPF varies from less than 1% to greater than 9% with a postoperative mortality rate of 25.7% [9]. Although grade B and C POPFs are both clinically relevant, grade C POPF is more significant due to its high mortality rate. However, while there have been several reports describing risk factors for grade B or C POPF, few studies have focused on grade C. The aim of this study was to discuss recent findings and evidence to analyze the prevention and treatment of patients with grade C POPF.

## 2. Risk Factors for Grade C POPF

Although there are many studies on risk factors for POPF, few studies have examined and identified risk factors for the development of grade C POPF. The risk factor most consistently shown to be predictive of grade C POPF after pancreatic surgery is soft gland texture, while other factors, such as male sex, small pancreatic duct diameter <3 mm, BMI ≥25.0 kg/m^2^, preoperative serum albumin <3.0 g/mL, high intraoperative blood loss, and an operative time ≥480 min, were also reported [10,11,12,13]. To better evaluate the characteristics of the pancreas before surgery, imaging examinations, including retrorenal fat thickness, subcutaneous fat area, and total fat area, were found to be significant predictors of grade C pancreatic fistula [14].

Despite these possible risk factors, they are not particularly helpful for prevention or treatment because of the usually unchangeable features in clinical practice. Chiba et al., in a recent retrospective study, found that the detection of gram-negative rods within the first 7 postoperative days was a significant factor after multivariate analysis. Interestingly, this study also found that neither preoperative biliary drainage nor cholangitis were independent risk factors for grade C POPF [15]. Therefore, the relationship between infection and grade C POPF may be a novel and useful direction of study.

## 3. The Relationship between Infection and Grade C POPF

With the development of surgical techniques and perioperative management, the incidence of pancreatic fistula has gradually decreased, whereas the mechanism of pancreatic fistula remains unclear. It is important to recognize patients who are likely to develop life-threatening grade C POPF. Some studies suggest that drainage culture positivity is an independent predictor of grade B/C POPF and that the onset of grade B/C POPF may be triggered by infection [16,17]. However, not all patients with drainage fluid contamination develop grade C pancreatic fistula, and the exact mechanism of infection leading to pancreatic fistula remains obscure. Major sources of microorganisms include the biliary system, intestinal system, and catheter retrograde infections. The hypothesis is that different types of microorganisms or microorganisms from different sources play different roles in the occurrence and development of pancreatic fistula, and only microorganisms from specific origins cause grade C POPF.

### 3.1. Different Sources of Infection between PD and DP

Interestingly, the incidence of grade C POPF varies from pancreatoduodenectomy to distal pancreatectomy. Several studies discuss the relationship between different operations and pancreatic fistula, indicating that rates of POPF range from 3–15% after partial pancreatoduodenectomy (PD) to >30% after distal pancreatectomy (DP), but the rates of life-threatening POPF, namely, grade C pancreatic fistula, are higher in PD than DP [10,18,19,20]. The major difference between PD and DP is whether digestive tract reconstruction should be performed. Pancreatoduodenectomy requires the reconstruction of the gastrointestinal passage, biliary tract, and pancreatic outflow, which may result in the contamination of the abdominal cavity with microorganisms, and microorganism findings in drainage liquid are risk factors for POPF, which may explain the higher incidence of grade C POPF in PD. In cases of the leakage of the pancreaticojejunostomy or the biliary anastomosis, this intraoperative contamination, as well as a direct leakage-associated route for microorganisms from the gastrointestinal tract or bile duct into the abdomen, may explain the presence of microorganisms in PD. For the smaller amounts of microorganisms present in DP, the probable cause is bacterial migration from the digestive tract or catheter retrograde infection. In view of the varying incidence of grade C POPF between PD and DP, it is crucial to determine the influence of different sources of microorganisms on grade C POPF.

### 3.2. Hypothesis 1: Intestinal Microorganisms Develop Grade C POPF in PD

For PD, it is possible that two routes of bacterial contamination into drainage fluid in the early postoperative period were intraoperative bile juice spillage and leakage from the jejunum via the pancreatic jejunal anastomotic site [16]. Previous studies have concluded that preoperative biliary drainage (PBD) is associated with infectious complications such as PF after PD and that PBD is a risk factor for bile contamination [21]. In a subsequent study, Ohgi et al. compared patients with bacterobilia with patients without bacterobilia in 2016. In this study, they found that bacterobilia rather than PBD was a risk factor for grade B/C POPF, and microorganisms from the infected bile may be the source of the grade B/C POPFs [22]. However, these studies did not address grade C pancreatic fistula alone. A recent systematic review including a total of 28 studies found identical microorganisms in bile and infectious sources in 48% (interquartile range 34–59%) of cases. Furthermore, this study concluded that POPF, overall postoperative morbidity, and mortality were not significantly influenced by bacterobilia [23]. One possible reason for the potential discrepancy between the findings of these reports is that not all microorganisms present in bile are important, and only those found in the postoperative abdominal fluid play crucial roles in the development of CR-POPF.

Recently, the gut microbiota was found to play significant roles in postoperative complications such as POPF [24,25]. Several studies analyzed microorganisms in drainage fluid and found that the most frequently detected microorganisms in POPF patients after PD were *Enterococcus* spp., *Candida* spp., and *Escherichia coli*, whereas *Staphylococcus* spp. was the most frequently detected microorganism after DP [26,27]. Furthermore, in the microbiologic analysis, *Enterococcus faecium* and *Candida albicans* isolates in POPF cultures were associated with particularly poor outcomes, such as postoperative hemorrhage, sepsis, reoperation, and even death. Another retrospective study assessed the clinical characteristics and microbiological results of the drainage cultures of pancreaticoduodenectomy or distal pancreatectomy patients and found that Candida species were independent risk factors for grade C POPF and were strongly associated with severe complications such as pseudoaneurysms and postoperative hemorrhages after pancreaticoduodenectomy [28]. Candida species, which are resident microorganisms in the upper gastrointestinal tract, can remain in the infection site as a substitution [29]. Meanwhile, Enterobacterales were demonstrated to be more common after PD than after DP, and grade C POPF has a higher proportion of Enterobacterales than grade B pancreatic fistula [30].

The above studies found that intestinal microorganisms such as *Enterococcus faecium* and *Candida albicans* are more likely to cause grade C pancreatic fistulas than biliary microorganisms, and the main pathway by which intestinal microorganisms enter the abdominal cavity is pancreaticojejunostomy dehiscence in PD. (Figure 1A).

### 3.3. Hypothesis 2: Some Intestinal Microbes Enter the Abdominal Cavity by Bacterial Translocation

In PD, microorganisms enter the abdominal cavity mainly through pancreaticojejunostomy anastomosis. However, for a smaller proportion of grade C POPF in PD with no pancreaticojejunostomy dehiscence, bacterial translocation or catheter retrograde infection may be an explanation for intestinal microorganisms appearing in the drainage fluid.

Drains are frequently placed at the time of pancreatic surgery, and whether drains influence grade C POPF is under debate. Drains allow for the evacuation of blood, pancreatic juice, bile, and lymphatic fluid, and by monitoring the drainage fluid, clinicians can adjust the treatment strategy in a timely manner. However, prolonged drainage may increase the chances of retrograde infection. Interestingly, several studies have found that there was no significant difference in the incidence of grade C POPF and reoperation in a comparison with and without a drain [31,32]. This finding also suggests that retrograde bacteria could not result in grade C POPF and that the real pivotal microorganism may enter the abdominal cavity by bacterial translocation. (Figure 1B).

Bacterial translocation (BT) is the migration of gut microorganisms or their products across the intestinal mucosal barrier into extraintestinal sites such as the mesenteric lymph nodes and the portal system and then into the systemic circulation and the processes of the liver, spleen, lung, and other organs [33,34]. Meanwhile, BT is responsible for predisposing patients undergoing major abdominal surgery to infectious complications [35,36]. There are several mechanisms that prevent microorganism migration from the gastrointestinal tract. First, the microbiota, with its diversity, can effectively inhibit colonization by ingested microorganisms and the overgrowth of resident microorganisms normally present at low levels within the gut [37]. Second, intestinal movement contributes to barrier function by flushing out pathogenic microbes and their products. Intestinal dysmotility or intestinal obstruction leads to bacterial overgrowth in the intestinal lumen and the distention of the intestinal wall, which can lead to the disruption of the intestinal wall barrier and the development of inflammation [38]. Third, the mechanical barrier includes the mucus layer, intestinal epithelial cells, and tight junctions between them. In the mucus layer, microbes obtain nutrients from it, and the microbiota is involved in the maintenance of the mucus layer in turn. Fourth, the local immune system is composed of IgA secreted by mucosal B cells or plasma cells and other antimicrobial compounds. Patients undergoing major abdominal surgery have several risk factors for the disruption of the structural and functional barriers, which may induce BT. Aiming at the abovementioned mechanisms, rational perioperative management in patients undergoing pancreas surgery is necessary to reduce BT and prevent grade C POPF.

From the above discussion, compared with biliary microorganisms, patients with intestinal microorganisms detected in abdominal drainage fluid are more likely to develop grade C pancreatic fistula. The main pathways by which intestinal microorganisms enter the abdominal cavity include pancreaticojejunostomy anastomosis and BT. For the above mechanisms, therefore, various surgical techniques and perioperative management have been attempted to prevent grade C pancreatic fistula.

## 4. Operation Technique

### 4.1. Pancreaticojejunostomy Dehiscence Is Found in Grade C POPF after PD on Radiographic Examination

On radiographic examination, grade C POPF was found to have wider diameters of pancreaticojejunostomy dehiscence (Figure 2A) than grade A and grade B POPF after pancreaticoduodenectomy. In this study, most grade C pancreatic fistulas were combined with infection, which can be interpreted as dehiscent anastomosis [39]. In addition to the complete dehiscence of the pancreaticojejunostomy, the crevice of the pancreaticojejunostomy (Figure 2B) can also cause grade C POPF. Another study found that CT-imaging findings of fluid collection on the dorsal side of pancreaticojejunostomy and operative area bubble signs remained significant predictive factors of delayed hemorrhage in POPF patients [40]. Because postoperative hemorrhage is one of the leading causes of death in grade C POPF, these CT manifestations might also indicate the occurrence of grade CPOPF.

### 4.2. Reconstruction Methods

Confirming the best reconstruction method for pancreatic surgery has remained controversial up to now. Pancreaticogastrostomy (PG) and pancreaticojejunostomy (PJ) are the most common reconstruction methods among the several available techniques. Local inflammation due to the leakage of the pancreatic anastomosis is induced by activated pancreatic enzymes on the one hand and the simultaneous leakage of enteric microorganisms on the other hand. Keck et al. [41] reported the rate of CR-POPF in a multicenter randomized controlled trial with 440 patients. They concluded that the rate of grade B/C fistula after PG versus PJ was not different, but grade C pancreatic fistula was not discussed. However, they performed a perioperative secondary endpoint analysis and found that there was no significant difference in perioperative in-house mortality (PG vs. PJ, 6% vs. 5%, *p* = 0.963) or 90-day mortality (PG vs. PJ, 10% vs. 5%, *p* = 0.167). Recently, several meta-analyses have compared PG and PJ, and most concluded that there was no significant difference in the rate of CR-POPF [42,43,44,45,46,47,48]. Cheng et al. [43] demonstrated that PJ probably has little or no difference from PG in the risk of postoperative mortality (3.9% versus 4.8%; RR 0.84, 95% CI 0.53 to 1.34; moderate-quality evidence). However, a recent meta-analysis compared 971 patients who underwent PJ end-to-side duct-to-mucosa, 791 patients who underwent PJ end-to-side invagination, 505 patients who underwent PG end-to-side invagination, and 98 patients who underwent PG end-to-side duct-to-mucosa and demonstrated that duct-to-mucosa pancreaticogastrostomy was associated with the lowest rates of clinically relevant POPF and had the best outcome profile, in terms of postoperative morbidity and postoperative mortality, among all pancreatic anastomosis techniques following PD [44]. Controversially, an RCT published in 2020 found that PJ had a lower incidence of grade C POPF than PG among patients at the highest risk for POPF (0% versus 11.1%) [49]. In contrast, this study applied a risk stratification system and included only patients at high risk for POPF, and an external stent was used in all patients.

Reconstruction after pancreatic surgery remains under debate, and most studies have found no difference in the rate of CR-POPF after PD. Therefore, the reconstruction method should be identified based on the condition of the patients, tumor characteristics, and the habits of surgeons.

### 4.3. Anastomotic Techniques

In addition to bowel reconstruction, anastomotic techniques are critical in pancreatic surgery. To date, there have been more than 50 different pancreaticojejunostomy methods, including end-to-end anastomosis and end-to-side anastomosis, coating-in, binding, and duct-to-mucosa anastomosis. The incidence of pancreatic fistula varies with different anastomosis methods. Currently, end-to-end invagination pancreaticojejunostomy and end-to-side duct-to-mucosa pancreaticojejunostomy are the two most widely used anastomosis methods in clinical practice. End-to-side duct-to-mucosa pancreaticojejunostomy includes anastomosis of the pancreatic duct and jejunum mucosa and anastomosis of pancreatic palpation and the jejunum plasma muscle layer, usually in the form of end-to-side anastomosis. However, the anastomotic procedure is not always easy, particularly with narrow pancreatic ducts and a soft pancreas. End-to-end invagination pancreaticojejunostomy, an easier reconstruction method, refers to the insertion of the intact pancreatic stump into the jejunum, which can be a single or double anastomosis.

A 120-patient RCT conducted by Senda et al. in 2018 found no significant difference in the incidence of clinical pancreatic fistulas between duct-to-mucosa PJ and invagination PJ, but in high-risk patients with a soft pancreas, the incidence of clinical pancreatic fistulas was better reduced by invagination PJ, and patients who underwent invagination PJ had shorter hospital stays [50]. The possible reason is that a soft pancreas is usually accompanied by a small main pancreatic duct diameter, which directly increases the difficulty of anastomosis between the pancreatic duct and jejunal mucosa. Some meta-analyses also demonstrated that invagination PJ did not reduce CR-POPF rates compared to duct-to-mucosa PJ, especially in patients with a hard pancreas [46,51]. However, a meta-analysis found a superiority of invagination anastomosis over duct-to-mucosa anastomosis in reducing the risk of grade B/C POPF, but it did not significantly reduce the mortality rate or length of hospital stay. The effect of invagination in reducing pancreatic fistula formation is obvious in patients with a soft pancreas, but there is no significant difference between the two anastomosis techniques in patients with a hard pancreas [52].

However, several recent studies have found a superiority of an uncommon anastomosis technology named Blumgart anastomosis (BA). BA was devised by LH Blumgart in 2000 and combines the principle of duct-to-mucosa anastomosis with the invagination technique of the pancreas remnant, showing a low rate of grade C POPF [53]. The original Blumgart technique was subsequently modified by decreasing the number of transpancreatic sutures and by tying the sutures only once over the anterior aspect of the jejunal limb, instead of first tying them on the pancreas before placing the anterior seromuscular suture on the jejunum [54]. Several studies found that BA reduced severe complications, such as POPF grade C, reoperation rate, and 90-day mortality, after PD compared to traditional PJ [54,55]. Therefore, BA could be used with more frequency, and the transpancreatic U-suture, the small incision of the jejunal loop, and jejunal wrapping seemed to be the key technical factors.

Similar to reconstruction methods, traditional anastomotic techniques after pancreatic surgery are under debate, and the results of most studies could be affected by many factors, including the condition of patients, characteristics of tumors, and the technology of surgeons. With the improvement of anastomosis technology, the new method of BA anastomosis may be reliable and feasible in clinical practice. Therefore, more high-quality RCTs are needed to confirm this notion.

### 4.4. Stent or No-Stent

Stent placement in PJ anastomosis is also controversial, as is whether the stent should be an external or internal stent. Several recent meta-analyses have found that stent use has the potential benefit of reducing the incidence of POPF [56,57,58]. However, Pessaux et al. found that stents mainly reduced the occurrence of grade B pancreatic fistula, but there was no significant difference in the incidence of grade C pancreatic fistula [59]. Compared with stents, external stents exert a more pronounced effect on the prevention of CR-POPF, especially grade C POPF [60,61]. The reason might be that the external stent minimizes the digestive erosion of the residual pancreas and fully eliminates pancreatic juices from the anastomosis.

From what has been discussed above, external stent application in PD is better than internal stent application in reducing the incidence of grade C POPF; however, whether stent use could reduce grade C POPF is still unclear. Because few studies have discussed grade C POPF alone, along with the superiority of external stents compared to internal stents, additional RCTs comparing the efficacy toward grade C POPF of external pancreatic duct stenting versus non-stenting must be performed, particularly for cases with a high risk for pancreatic fistula.

### 4.5. Two-Stage Surgical Procedure

A two-stage surgical procedure was undertaken to prevent the development of POPF in patients with high-risk pancreases, which contained two approaches: completion pancreatectomy or two-stage pancreaticojejunostomy. The purpose of the planned PD intraoperatively converted to completion pancreatectomy is to remove the source of inflammation and leakage and to prevent POPF. Recently, several analyses have demonstrated that completion pancreatectomy exhibited lower rates of major morbidity and comparable mortality compared to high-risk PD [62,63,64]. Despite acceptable postoperative outcomes, performing completion pancreatectomy still raises important concerns due to the inevitable presence of endocrine and exocrine insufficiency. Since completion pancreatectomy does not appear to reduce mortality, completion pancreatectomy may be considered rather than high-risk PD only in a few selected cases. Another method is two-stage pancreaticojejunostomy. In the two-stage PJ method, the insertion of an external tube is performed first, in which the tube is not passed through the jejunal loop, followed approximately 3 months later by second-stage reconstruction for PJ. A few studies have reported this method with a low rate of postoperative hemorrhage and mortality and indicated that two-stage PJ may be a safe and useful method in PD in patients with a soft pancreas or a narrow main pancreatic duct [65,66,67]. However, the availability of both methods requires more high-quality studies in the future.

## 5. Perioperative Management According to BT Theory

### 5.1. Mechanical Bowel Preparation Disturbs the Intestinal Mucosal Immune Barrier

Preoperative intestinal preparation includes mechanical intestinal preparation and oral antibiotics to remove intestinal microorganisms. Mechanical bowel preparation (MBP) was defined as taking sodium phosphate or polyethylene glycol electrolyte powder orally before the operation to clear the bowel. However, mechanical bowel preparation before pancreatic surgery is questionable and controversial in modern perioperative care. Because of a large incidence of diarrhea, mechanical intestinal preparation can lead to dehydration and electrolyte disturbances, especially in elderly patients [68]. Recently, more studies have found that no evidence was noted supporting the use of MBP in patients undergoing elective colorectal surgery, and MBP should be omitted in routine clinical practice [69,70]. Additionally, Toh et al. reported that mechanical bowel cleaning alone is not helpful in reducing infectious complications such as surgical site infection (SSI) compared to oral antibiotics and bowel cleaning [71]. Meanwhile, a large retrospective analysis has shown that mechanical intestinal preparation before pancreaticoduodenal surgery does not benefit patients [72]. In pancreas surgery, MBP may result in the disturbance of the intestinal flora and damage to the bowel mechanical barrier, which promote BT and increase the incidence of grade C POPF. Hence, mechanical bowel preparation should not be administered routinely prior to pancreatic surgery.

### 5.2. Avoid Excessive Fluid after Surgery to Reduce Intestinal Edema

The main goals of postoperative fluid therapy in attempting to limit BT include maintaining intestinal perfusion to limit ischemia—reperfusion injury and ensuring the continuity of the epithelial barrier and mucus layer, according to the three-hit model theory of BT [73]. However, too much fluid for infusion could also result in BT by causing intestinal edema and increasing gut permeability. Optimal perioperative fluid management for major intraabdominal operations remains under debate, and there are several different fluid management strategies in the clinic. Goal-directed fluid therapy (GDFT) was proposed by introducing different hemodynamic variables into a dynamic perspective of individual fluid loading with or without vasoactive substances to reach a predefined goal of optimal preload and oxygen delivery [74]. Several studies have demonstrated the superiority of GDFT in major abdominal operations by reducing surgical complications [75,76,77,78,79]. Yuan et al. analyzed the results of 29 studies and showed that GDFT significantly reduced the incidence of SSIs after abdominal surgery, which is associated with grade C POPF, as well as the length of hospital stay [75]. In pancreatic surgery, a goal-directed strategy for the management of perioperative fluids can also result in fewer complications, including pancreatic fistula [77]. On the one hand, rational fluid therapy could reduce intestinal tension, preventing anastomotic dehiscence; on the other hand, optimal fluid could reduce intestinal edema and microorganism translocation. For intraoperative fluid therapy, a meta-analysis indicated that there are no significant differences between restrictive fluid therapy and goal-directed fluid therapy in a low-risk population undergoing abdominal surgery [80]. Another meta-analysis demonstrated that intraoperative fluid restriction in PD did not significantly affect postoperative outcomes [81].

Despite fewer studies about the relationship between GDFT and grade C POPF directly, many studies have demonstrated that postoperative GDFT could reduce the length of delayed gastric emptying and infective complications associated with grade C POPF. In conclusion, for patients undergoing pancreatic surgery, postoperative GDFT may be beneficial for decreasing the incidence of grade C POPF and other complications.

### 5.3. Oral Nutrition Is Beneficial for Maintaining Intestinal Mucosal Immunity

PD results in a loss of gastric pacemakers and partial pancreatic resection, leading to a high incidence of postoperative malnutrition. Surgical trauma could increase immune system suppression, and surgical injury as well as malnutrition can induce poor postoperative outcomes. Malnutrition is worse in the early postoperative days and affects the process of healing, intestinal barrier function, and the incidence of postoperative complications, especially POPF [82,83]. Therefore, postoperative nutritional support, including early enteral nutrition (EEN) and total parenteral nutrition (TPN), is an important method of improving the patient’s nutritional status and preventing postoperative complications, as recommended by the guidelines of the European Society for Clinical Nutrition and Metabolism [84]. However, the efficacy and safety of EEN and TPN after PD are under debate. Several analyses have found that EN is beneficial in reducing infectious complications and the length of hospital stay in patients after PD [85,86,87]. The hypothesis that EN is beneficial in reducing infectious complications is supported by evidence that modifications in mucosal defense have been implicated as important factors affecting infectious complications in critically ill patients and that EN influences the ability of gut-associated lymphoid tissue to maintain mucosal immunity [88]. However, another analysis demonstrated that TPN is superior to EN in reducing the incidence of DGE and pneumonia, shortening nasogastric tube retention, and decreasing hospitalization expenses [89]. Furthermore, Perinel et al. even found that nasojejunal early enteral nutrition was associated with an increased overall postoperative complication rate in patients undergoing PD, and the frequency and severity of POPF were significantly increased after nasojejunal early enteral nutrition [90]. Therefore, postoperative nasojejunal EN should be given only on selective indications rather than routinely used, such as for patients with a severe malnourished status [91]. Another study compared five feeding routes after pancreatoduodenectomy and found that the oral diet is superior to other feeding routes [92]. Therefore, among patients with high-risk POPF, EN, and especially oral nutrition, may theoretically be a feasible method for maintaining intestinal flora structure and reducing the peripheral fluid volume to avoid grade C POPF, whereas early nasojejunal EN is not recommended for routine use. In addition, some patients cannot tolerate enteral nutrition (EN) owing to diarrhea, gastric spasms, and other symptoms, while parenteral nutrition may be an optional method for these patients.

Although the determinant of pancreatic fistulas is still whether PJ anastomosis is dehiscent, rational perioperative management is necessary to reduce grade C POPF. On the one hand, applicable perioperative management could protect PJ anastomosis indirectly. On the other hand, it could reduce BT effectively with multiple mechanisms. By reviewing previous articles, nonmechanical intestinal preparation, GDFT, and appropriate early enteral nutrition may have a potential positive effect on reducing grade C POPF. Somatostatin analogs as a potential measure for POPF are under debate and a majority of studies have demonstrated that somatostatin analogs have less effect on POPF [93,94,95]. Other measures, such as probiotics could reinforce the host defense system to limit BT, but their effect on grade C POPF remains unclear [96,97]. Moreover, we summarize the possible causes of grade C postoperative pancreatic fistula and the corresponding preventive measures in this review (Table 1).

## 6. Surgical Treatment for Grade C POPF

According to the definition revised by ISGPF, grade C POPF usually requires surgical treatment. During relaparotomy, different surgical strategies are possible: completion pancreatectomy, the revision of the pancreatic anastomosis, and the bridge stent technique [98].

### 6.1. Debridement and Open Drainage

For POPF, a step-up approach is usually used in fluid collection, which is first treated by antibiotic therapy and then radiologic-guided drainage or endoscopic ultrasound-guided drainage. If the patients developed postoperative abdominal bleeding, hemostatic therapy or angiographic procedures were needed. Once conservative treatment fails, reoperation becomes the last option. Debridement and drainage are the basic operations for patients with grade C pancreatic fistula after the failure of interventional procedures. The advantages of this approach are a short operation time, less trauma, and safety, and simple drainage is significantly superior in most evaluated parameters, such as the lower rate of 90-day mortality. However, it is frequently associated with high rates of reoperations compared to completion pancreatectomy and PJ disconnection [99,100]. Therefore, debridement and drainage seem to be indicated in patients with uncomplicated dehiscence or severe hemodynamic instability.

### 6.2. Completion Pancreatectomy

Completion pancreatectomy is the most aggressive strategy to treat grade C POPF, especially in cases of PJ dehiscence with severe inflammation. A theoretical advantage of completion pancreatectomy is that it completely removes the focus of intra-abdominal leakage and the source of inflammation, thereby possibly decreasing the risk of additional relaparotomies. However, this radical surgical strategy may induce the deterioration of organ failure, intraoperative blood loss, the risk of damaging other structures, and pancreatic exocrine and endocrine insufficiency [101]. Due to the high mortality and the low quality of life of patients after completion pancreatectomy, surgeons prefer a pancreas-preserving procedure to completion pancreatectomy during relaparotomy, and only in the case of pancreatic necrosis or the complete destruction of the former anastomosis can completion pancreatectomy be reserved. According to the above theories of intestinal fluid and intestinal bacteria in the occurrence of grade C POPF, another surgical treatment was gradually applied that blocked the flow of pancreatic fluid to the intestines to control local infection.

### 6.3. Revision of the Pancreatic Anastomosis

Repeating PJ anastomosis usually includes two parts: salvage relaparotomy and wirsungostomy, and delayed restorative laparotomy reconstructing the PJ anastomosis. A study implemented this strategy in patients with grade C POPF, and most patients underwent repeat pancreaticojejunostomy after wirsungostomy with unaltered long-term endocrine function [102]. The precondition of repeating PJ anastomosis is that the patients survive after wirsungostomy. Several analyses have found that wirsungostomy seems not to be inferior to open drainage procedures or completion pancreatectomy in terms of mortality rate [99,103]. Meanwhile, this treatment for grade C POPF has the advantage of preserving pancreatic function compared to completion pancreatectomy and may be an alternate method for surgeons.

### 6.4. The Bridge Stent Technique

The bridge stent technique is an alternative approach for managing PJ dehiscence in the following steps: a. Transpancreatic U-stitches are placed at the edge of the pancreatic stump to avoid leakage; b. The distal part of the thin silicon tube stenting the pancreatic duct is inserted into the jejunal lumen and fixed by purse-string sutures to the jejunal wall; c. An irrigating catheter and drains are put in place [104]. Kent et al. used this technique in five patients with grade C POPF, and all those patients survived this complication and were discharged from the hospital without long-term complications of PJ anastomosis [105]. Therefore, the bridge stent technique may be a safe and effective method for treating grade C POPF while retaining pancreatic function.

Grade C POPF following pancreaticoduodenectomy is a rare and difficult problem to manage. Traditionally, re-exploration with completion pancreatectomy and wide drainage has been employed, but this process carries significant morbidity and mortality. The revision of the pancreatic anastomosis or the bridge stent technique seems to have a higher priority than completion pancreatectomy, and studies with a larger sample size and longer follow-up are required to establish the short- and long-term efficacy.

## 7. Discussion

Different sources of infection have different influences on the development of pancreatic fistula, and grade C POPF is mainly caused by intestinal sources of infection. The dehiscence of PJ anastomosis and BT are the two main pathways by which intestinal microorganisms enter the abdominal cavity. For operative techniques, the condition of the patients, tumor characteristics, and the habits of surgeons are important factors in deciding the surgical strategy. BA anastomosis may be reliable and feasible in clinical practice, and the effectiveness of external stents in preventing grade C POPF needs further validation. For perioperative management, avoiding mechanical bowel preparation and excessive intravenous fluids as well as early enteral nutrition are important methods of avoiding BT. The main problem with pancreatic fistula is late bleeding, which is the primary reason for mortality. Thus, to be prepared for dealing with bleeding, interventional radiology must be available 24 h a day, 365 days per year. At present, the step-up approach is mainly applied in acute pancreatitis. Step-up approaches in acute pancreatitis include percutaneous drainage, endoscopic (transgastric) drainage, and minimally invasive retroperitoneal necrosectomy and are increasingly being used. Compared to open necrosectomy, the step-up approach reduces the rate of major complications and death [106,107]. In POPF, there are also applications for the step-up approach. Smits et al. suggest a beneficial effect of catheter drainage instead of primary relaparotomy on morbidity and mortality. However, even in a step-up approach, relaparotomy will sometimes be unavoidable in some patients, and the delay of surgery usually results in a worse prognosis [62,108]. Meanwhile, some predictive factors, such as postoperative computerized tomography evidence of pancreaticojejunostomy dehiscence and operative area bubble signs, may indicate death in pancreatic fistula-associated hemorrhage patients who underwent pancreatoduodenectomy [109]. Therefore, it is also important to investigate the predictors for grade C POPF that can support early decision-making regarding the value of a relaparotomy in vulnerable, high-risk patients, and to analyze the timing of relaparotomy.

## Figures and Tables

**Figure 1 jcm-11-07516-f001:**
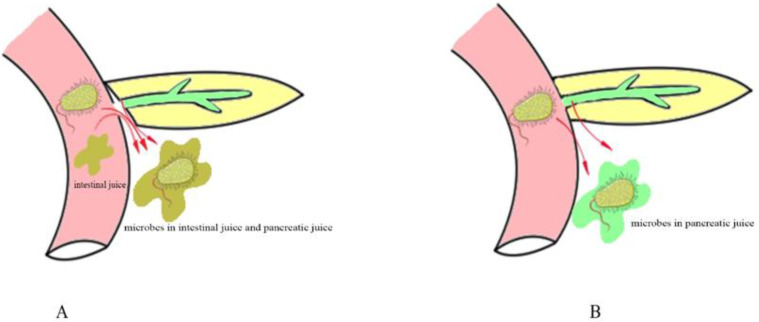
Different pathways of intestinal microbial leakage in PD. (**A**) shows that microbes leak from the pancreaticojejunostomy anastomosis. (**B**) shows that microbes leak by bacterial translocation.

**Figure 2 jcm-11-07516-f002:**
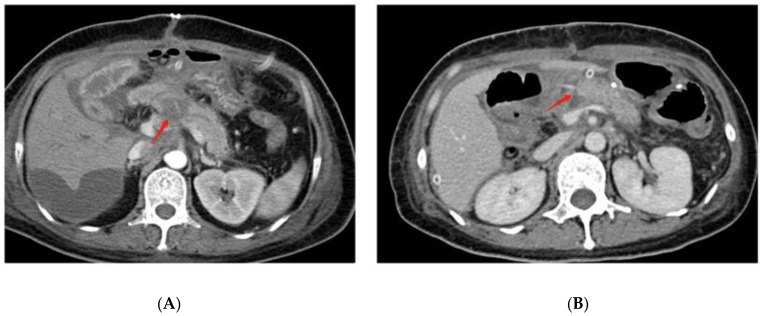
Postoperative CT findings of grade C POPF. (**A**) shows the complete dehiscence of the pancreaticojejunostomy at the red arrow. (**B**) shows the crevice of the pancreaticojejunostomy at the red arrow.

**Table 1 jcm-11-07516-t001:** Possible causes of grade C postoperative pancreatic fistula and corresponding preventive measures.

Possible Causes of Grade C Postoperative Pancreatic Fistula		Preventive Measures
Intestinal juice spillover		Reconstruction methods
		Anastomotic techniques
		Stent or no-stent
Bacterial translocation	Mucosal barrier breakdown	Avoid mechanical bowel preparation
		Oral enteral nutrition
	Edema of intestinal wall	Goal-directed fluid therapy

## Data Availability

No new data were created or analyzed in this study. Data sharing is not applicable to this article.

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
