# Peer review of "Prevention and Treatment of Grade C Postoperative Pancreatic Fistula"

_jcm, 2022, doi:10.3390/jcm11247516_

Round 1

Reviewer 1 Report

Authors in this review discusses the two main pathways of grade C POPF, direct intestinal juice spillover and bacterial translocation, by which intestinal microbes enter the abdominal cavity and based on these theories, they summarize the operation techniques and perioperative management of grade C POPF  and discussed the novel methods and surgical treatment to reverse this dilemma. Although well written still require different comments

1. Abstract line 4-5. Please correct the sentence "...However, the only measure is reoperation for this fatal postoperative complication based on expert consensus with a poor prognosis..." . Reoperation is not the only measure for this complication. Indeed reoperation is the last change against pancreatic fistula with high morbidity and mortality. Conservative management is the real strategy in those pt. Please rewrite the sentence

2. Chapter 6. Completion of pancreatectomy is the most aggressive strategy with high mortality and  must be reserved only in the case of pancreatic necrosis or complete destruction of the former anastomosis and is not recommended in very critical ill pts. In this pts is better a Wirsungostomy and drainage

3. Indeed no real definitive recomandation could be made but vigilant behavior may enhance outcomes

4. Please discuss the "step-up approach" in the POPF treatment

5. Plaese include a section of somatostatine analogues in the prevention of POPF

6. The sentence about "revision of the PJ" is not correct : "..This treatment for grade C POPF has the advantage of preserving pancreatic function and reducing complications compared to complete pancreatectomy, and it  is the preferred choice in clinical practice..". Indeed in only very few cases a redo anastomosis must be done and is not the preferred choice at all 

7. In the reality the best treatment is to keep cool and do not overreact, since 90% of fistulas will heal by just healing. The main problem of fistula is late bleeding; this is the primary reason for mortality. Thus, be prepared for dealing with bleeding; interventionalradiology must be available 24 hours a day, 365 days per year. Please add this sentence 

8. No mention to the conservative POPF management in reported. Please add a separate chapter

Author Response

Response letter

We thank you for the valuable and constructive suggestions, which led to significant improvements of the quality of our manuscript. In following pages, there are our point-by-point responses to each of the comments.

Reviewer:

Authors in this review discusses the two main pathways of grade C POPF, direct intestinal juice spillover and bacterial translocation, by which intestinal microbes enter the abdominal cavity and based on these theories, they summarize the operation techniques and perioperative management of grade C POPF and discussed the novel methods and surgical treatment to reverse this dilemma. Although well written still require different comments.

  1. Abstract line 4-5. Please correct the sentence "...However, the only measure is reoperation for this fatal postoperative complication based on expert consensus with a poor prognosis..." . Reoperation is not the only measure for this complication. Indeed reoperation is the last change against pancreatic fistula with high morbidity and mortality. Conservative management is the real strategy in those pt. Please rewrite the sentence

Response: Thank you for your valuable suggestions. We have rewritten that sentence as “The patients with grade C POPF finally undergo surgery with a poor prognosis after various failed conservative treatment.”.

  1. Chapter 6. Completion of pancreatectomy is the most aggressive strategy with high mortality and must be reserved only in the case of pancreatic necrosis or complete destruction of the former anastomosis and is not recommended in very critical ill pts. In this pts is better a Wirsungostomy and drainage

Response: We agree with the reviewer that completion of pancreatectomy was usually reserved in the case of pancreatic necrosis or complete destruction of the former anastomosis and apology for the lack of clarity in this section. Therefore, after careful consideration, we have added that “Due to the high mortality and the low quality of life of patients after completion pancreatectomy, surgeons prefer a pancreas-preserving procedure to completion pancreatectomy during relaparotomy and only in the case of pancreatic necrosis or complete destruction of the former anastomosis can completion pancreatectomy be reserved. According to the above theories of intestinal fluid and intestinal bacteria in the occurrence of grade C POPF, another surgical treatment was gradually applied which block the flow of pancreatic fluid to the intestines to control local infection.”

  1. Indeed, no real definitive recommendation could be made, but vigilant behavior, may enhance outcomes

Response: Thank you for your valuable suggestions. At present, there is no definitive recommendation on the treatment of grade C pancreatic fistula. The current studies towards grade C POPF are heterogeneous and different patients may need different treatment. It is important to identify POPF early and perform necessary treatment.

  1. Please discuss the "step-up approach" in the POPF treatment

Response: Thanks for your valuable suggestion. we have added that section in the discussion. “At present, step-up approach is mainly applied in acute pancreatitis. Step-up approach in acute pancreatitis include percutaneous drainage, endoscopic (transgastric) drainage, and minimally invasive retroperitoneal necrosectomy, and were increasingly being used. Compared to open necrosectomy, step-up approach reduced the rate of major complications and death. In POPF, there are also applications for step-up approach. Smits et al. suggest a beneficial effect of catheter drainage instead of primary relaparotomy on morbidity and mortality. However, even in a step-up approach, relaparotomy sometimes will be unavoidable in some patients and delay of surgery usually resulted in a worse prognosis. Meanwhile, some predictive factors like postoperative computerized tomography evidence of pancreaticojejunostomy dehiscence and operative area bubble signs may indicate death in pancreatic fistula-associated hemorrhage patients who underwent pancreatoduodenectomy. Therefore, it is also important to investigate the predictors for grade C POPF that can support the early decision-making regarding the value of a relaparotomy in vulnerable, high-risk patients and to analysis the timing of relaparotomy.”

  1. Plaese include a section of somatostatine analogues in the prevention of POPF

Response: Thanks for your suggestions. We have already discussed the use of somatostatine analogues in POPF in lines 409 to 411 of the manuscript. Due to the limited evidence of a beneficial effect of somatostatine analogues on grade C POPF, we gave this section only a few pages.

  1. The sentence about "revision of the PJ" is not correct : "..This treatment for grade C POPF has the advantage of preserving pancreatic function and reducing complications compared to complete pancreatectomy, and it is the preferred choice in clinical practice..". Indeed in only very few cases a redo anastomosis must be done and is not the preferred choice at all

Response: Thanks for your suggestions. In this section we discussed wirsungostomy and repeating PJ anastomosis simultaneously. We have added the relevant content as "The precondition of repeating PJ anastomosis is that the patients survive after wirsungostomy. Several analyses have found that wirsungostomy seems not to be inferior to open drainage procedure or completion pancreatectomy regarding the rate of mortality rate. Meanwhile, this treatment for grade C POPF has the advantage of preserving pancreatic function compared to complete pancreatectomy and may be an alternate method by the surgeons. "

  1. In the reality the best treatment is to keep cool and do not overreact, since 90% of fistulas will heal by just healing. The main problem of fistula is late bleeding; this is the primary reason for mortality. Thus, be prepared for dealing with bleeding; interventionalradiology must be available 24 hours a day, 365 days per year. Please add this sentence

Response: Thanks again for your valuable suggestion. We revisit the subject and have added in the discussion as “The main problem of pancreatic fistula is late bleeding and this is the primary reason for mortality. Thus, to be prepared for dealing with bleeding, interventional radiology must be available 24 hours a day, 365 days per year.”

  1. No mention to the conservative POPF management in reported. Please add a separate chapter

Response: Thanks for your suggestion. We have added a new paragraph and mention this part in “6.1 Debridement and drainage”.

Reviewer 2 Report

This study was aimed to review the issues of pathogenesis and therapeutic strategies of grade C postoperative pancreatic fistula after pancreatectomy, especially in patients undergoing pancreaticoduodenectomy/

1, This study was prepared with a narrative reviewing style, not with systematic review and or metanalysis. Therefore, the review seemed to be very subjective sense. The pathogenetic description of the occurrence of postoperative grade C pancreatic fistulae after pancreaticoduodenectomy was considered by two different pathogenetic infectious scenario. The first process was direct dehiscence of pancreatico-enterostomy. The other process was bacterial translocation. From these points of view, authors reviewed the prophylactic and therapeutic strategies that have been investigated before. Their consideration might be very unique and interesting for readers.

2, The one important point of preventive strategies of grade C postoperative pancreatic fistulae following pancreaticoduodenectomy is two-staged pancreaticoduodenectomy. Authors should describe about this issue in the session of “ Perioperative management”.

3, In session “ Completion pancreatectomy” in page 9, authors described that “due to the low quality of life of patients after completion pancreatectomy, completion pancreatectomy is probably the surgical option for crucially ill patients or patients after other interventions” It might be a justified opinion that that total completion pancreatectomy is a finally selected option for controlling grade C pancreatic fistulae after utilizing other interventional therapeutic options. However, patients undergoing total pancreatectomy could be doing well conditioned life as compared with previous time at the present time.

Therefore, this sentence should be appropriately revised about this point.

Author Response

Response letter

We thank you for the valuable and constructive suggestions, which led to significant improvements of the quality of our manuscript. In following pages, there are our point-by-point responses to each of the comments.

Reviewer:

This study was aimed to review the issues of pathogenesis and therapeutic strategies of grade C postoperative pancreatic fistula after pancreatectomy, especially in patients undergoing pancreaticoduodenectomy/

1, This study was prepared with a narrative reviewing style, not with systematic review and or metanalysis. Therefore, the review seemed to be very subjective sense. The pathogenetic description of the occurrence of postoperative grade C pancreatic fistulae after pancreaticoduodenectomy was considered by two different pathogenetic infectious scenario. The first process was direct dehiscence of pancreatico-enterostomy. The other process was bacterial translocation. From these points of view, authors reviewed the prophylactic and therapeutic strategies that have been investigated before. Their consideration might be very unique and interesting for readers.

Response: Thank you for your valuable comments. This review discussed the role of the intestinal microbiome in grade C POPF and related perioperative management from these points of view. Intestinal fluid or intestinal flora may play a facilitating role in POPF which is the core view in this review.

2, The one important point of preventive strategies of grade C postoperative pancreatic fistulae following pancreaticoduodenectomy is two-staged pancreaticoduodenectomy. Auth ors should describe about this issue in the session of “ Perioperative management”.

Response: Thanks for your suggestions. After reviewing related literature, we have added this part in the section “4.5 two-stage surgical procedure”.

3, In session “ Completion pancreatectomy” in page 9, authors described that “due to the low quality of life of patients after completion pancreatectomy, completion pancreatectomy is probably the surgical option for crucially ill patients or patients after other interventions” It might be a justified opinion that that total completion pancreatectomy is a finally selected option for controlling grade C pancreatic fistulae after utilizing other interventional therapeutic options. However, patients undergoing total pancreatectomy could be doing well conditioned life as compared with previous time at the present time.

Therefore, this sentence should be appropriately revised about this point.

Response: Thanks again for your valuable suggestions. The current utilization rate of TP is increasing, and the short-term postoperative outcome is superior to that of in high-risk PD. Due to the standardized management of endocrine and exocrine insufficiency in pancreas, the influence of TP on quality of life is gradually diminishing. However, since TP does not reduce postoperative mortality compared to other methods, TP could only be considered as an alternative to patients with pancreatic necrosis or complete destruction of the former anastomosis. Therefore, we have revised this sentence as “Due to the high mortality and the low quality of life of patients after completion pancreatectomy, surgeons prefer a pancreas-preserving procedure to completion pancreatectomy during relaparotomy and only in the case of pancreatic necrosis or complete destruction of the former anastomosis can completion pancreatectomy be reserved.”.

Round 2

Reviewer 1 Report

In the revised version Authors made the requested changes

Author Response

Dear reviewer:

Thanks very much for taking your time to review this revised manuscript. We really appreciate all your generous comments and suggestions!